# A Novel Tree Biomass Estimation Model Applying the Pipe Model Theory and Adaptable to UAV-Derived Canopy Height Models

**Takashi Machimura [1,\*], Ayana Fujimoto [1], Kiichiro Hayashi [2], Hiroaki Takagi [3] and Satoru Sugita [4]**

1. Graduate School of Engineering, Osaka University, Yamadaoka 2-1, Suita, Osaka 565-0871, Japan; pikacookie0603@gmail.com
2. Institute of Materials and Systems for Sustainability, Nagoya University, Furo-cho, Chikusa, Nagoya 464-8601, Japan; maruhaya98–@nagoya-u.jp
3. Department of Civil Engineering, Nagoya University, Furo-cho, Chikusa, Nagoya 464-8601, Japan; 89dr760711t2h498@gmail.com
4. International Digital Earth Applied Science Research Center, Matsumoto-cho 1200, Kasugai 487-8501, Japan; satoru@isc.chubu.ac.jp
* Correspondence: mach@see.eng.osaka-u.ac.jp; Tel.: +81-6-6879-7391

**Abstract:** Aiming to develop a new tree biomass estimation model that is adaptable to airborne observations of forest canopies by unmanned aerial vehicles (UAVs), we applied two theories of plant form; the pipe model theory (PMT) and the statical model of plant form as an extension of the PMT for tall trees. Based on these theories, tree biomass was formulated using an individual tree canopy height model derived from a UAV. The advantage of this model is that it does not depend on diameter at breast height which is difficult to observe using remote-sensing techniques. We also proposed a treetop detection method based on the fractal geometry of the crown and stand. Comparing surveys in plantations of Japanese cedar (*Cryptomeria japonica* D. Don) and Japanese cypress (*Chamaecyparis obtusa* Endl.) in Japan, the root mean square error (RMSE) of the estimated stem volume was 0.26 $m^3$ and was smaller than or comparative to that of models using different methodologies. The significance of this model is that it contains only one empirical parameter to be adjusted which was found to be rather stable among different species and sites, suggesting the wide adaptability of the model. Finally, we demonstrated the potential applicability of the model to light detection and ranging (LiDAR) data which can provide vertical leaf density distribution.

**Keywords:** the statical model of plant form; Japanese cedar; Japanese cypress; treetop detection; fractal geometry

## 1. Introduction

Forests are essential for the conservation of biodiversity and soil and water resources as well as for providing forest products [1]. They can make significant contributions to the economy, livelihoods, and environment [2]. To promote the implementation of sustainable management of all types of forest has been stated as a target in Goal 15 (life on land) of the United Nations' Sustainable Development Goals (UN SDGs) [3], and the multiple functions and benefits of forests are directly and indirectly linked to various goals such as water and sanitation, sustainable energy, and climate change action [4]. Therefore, sustainable forest management has a significant impact on human and wildlife welfare.

Forest monitoring and inventory are fundamental activities for sustainable forest management that are indispensable for planning and validating management practices. Remote sensing is used in a large variety of forest monitoring, inventorying, and mapping [5] in both research and management applications. In recent years, unmanned aerial vehicles (UAVs) have been increasingly used owing to their low material and operational costs, and high-intensity data collection [5]. They are suitable for monitoring relatively small areas of

forest stands at high resolution and in short intervals. Generic commercial UAV systems carry a digital camera and are often utilized for observing 3D forest canopy structures by using the structure from motion (SfM) technique which generates surface points of objects from multiple photos [6–9]. Innovations in miniaturizing light detection and ranging (LiDAR) devices boarded on UAVs [10–12] have enabled more accurate observations of forest canopy structure than those using the SfM technique [13].

Tree biomass and its growth are fundamental indicators in forest management, and the allometric relationship among tree size metrics and tree mass or volume is widely used in surveys on the ground that utilize destructive observation methods. Stem volume is proportional to $D^2H$, that is, the squared diameter at breast height (DBH) multiplied by tree height, which is a standard formula in forestry practices. To estimate tree biomass using UAV-SfM or -LiDAR techniques, tree height is easily measurable but DBH has rarely been directly derived from the air, except in a few studies in open stands [14–16]. There are three alternative approaches free tree biomass estimation using airborne observations: estimating DBH and applying $D^2H$ or similar allometric relationships, estimating biomass from a non-linear multivariate regression of measurable crown metrics, and estimating biomass directly from LiDAR metrics with machine learning algorithms. In the first approach, crown width, crown area and tree height are used as predictors of DBH [17–19]; however, the accuracy of biomass or stem volume using estimated DBH has not been reported. In the second approach, crown diameter, crown area, crown surface area, and tree height are used as predictors of above ground biomass or stem volume [19–21]. The third approach utilizes a variety of LiDAR metrics such as height and density of return signals, as well as secondary metrics produced from signal statistics together with k-nearest neighbors and random forest algorithms [11,22]. These approaches for direct tree biomass estimation from remotely sensed data can potentially achieve high accuracy; however, overfitting, that is, the generalization inability of the optimized models and parameters to forests with varied species, ages, and growing environments, should be considered.

Shinozaki et al. found a general law of plant form and called it the pipe model theory (PMT) [23,24], in which, the amount of leaves existing above a certain level in a plant community is always proportional to the sum of the cross-sectional area of the stems and branches found at that level. For tall trees, this law is effective only in tree crowns; however, Oohata and Shinozaki later extended it to trunks below crowns and named it the statical model of plant form [25]. Although the validity of the PMT assessed by modern knowledge of biology is limited, especially from the perspective of the hydraulic properties of plants, it can still be a portfolio of a unified framework of plant function and structure [26]. We excavated this old theory because it is capable of calculating aboveground biomass from cumulative leaf mass from the treetop; therefore, it does not require an allometric relationship between stem volume and DBH. This is a significant benefit of the theory for tree biomass estimation from UAVs. Details of the formulization of theory to adapt it to UAV observations will be presented in the next section.

Based on the background and idea shown above, this study aimed to develop a new tree biomass estimation model applying the PMT, and to assess its validity in two coniferous plantations in Japan. We also improved the methodology of processing UAV-derived point clouds developed in our previous study [6], which included canopy height model (CHM) generation from point clouds, individual treetop detection and CHM correction for individual tree crowns.

## 2. New Tree Biomass Estimation Model Applying the Pipe Model Theory (PMT)

According to the PMT [23,24], cumulative leaf mass above any level in a crown is proportional to the total cross section of stems and branches that support the leaves at the same level. In the statical model of plant form [25] as an extension of the PMT to tall trees, an individual tree consists of crown and below-crown (trunk) parts (Figure 1), and similarly to inside a crown, the cumulative total (leaf and woody) mass above any level in a trunk is proportional to the cross section of the stem at the same level.

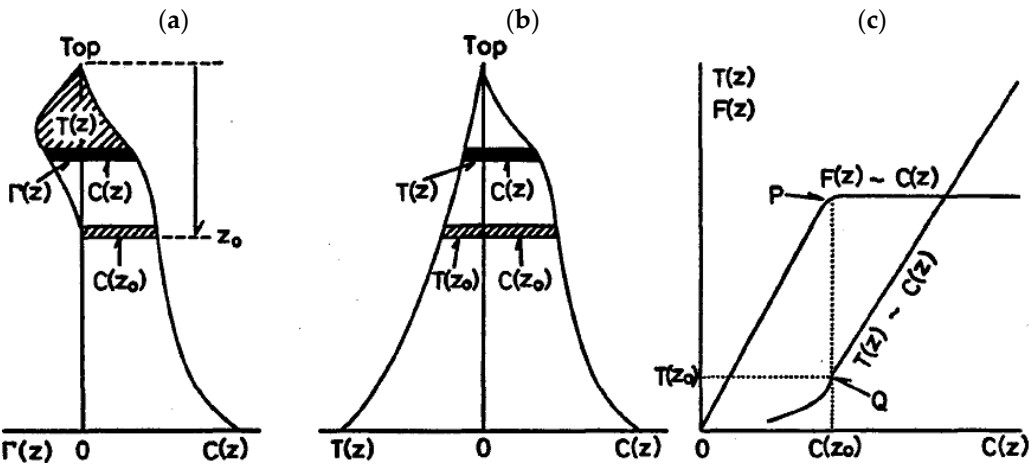

**Figure 1.** Concept of the statical model of plant form as an extension of the pipe model theory (PMT) to tall trees (Oohata and Shinozaki, 1979) [25] (Figure 2). (**a**) Vertical profiles of leaf and woody mass per length. (**b**) Vertical profiles of cumulative total (leaf and woody) mass from the top and woody mass per length. (**c**) Relationship between cumulative leaf/total mass from the top and woody mass per length. Γ and C are the leaf and woody mass per length at distance z from the treetop, respectively; F and T are the cumulative leaf and total mass from the treetop, respectively; and $z_0$ is the crown length at which the crown and trunk are divided.

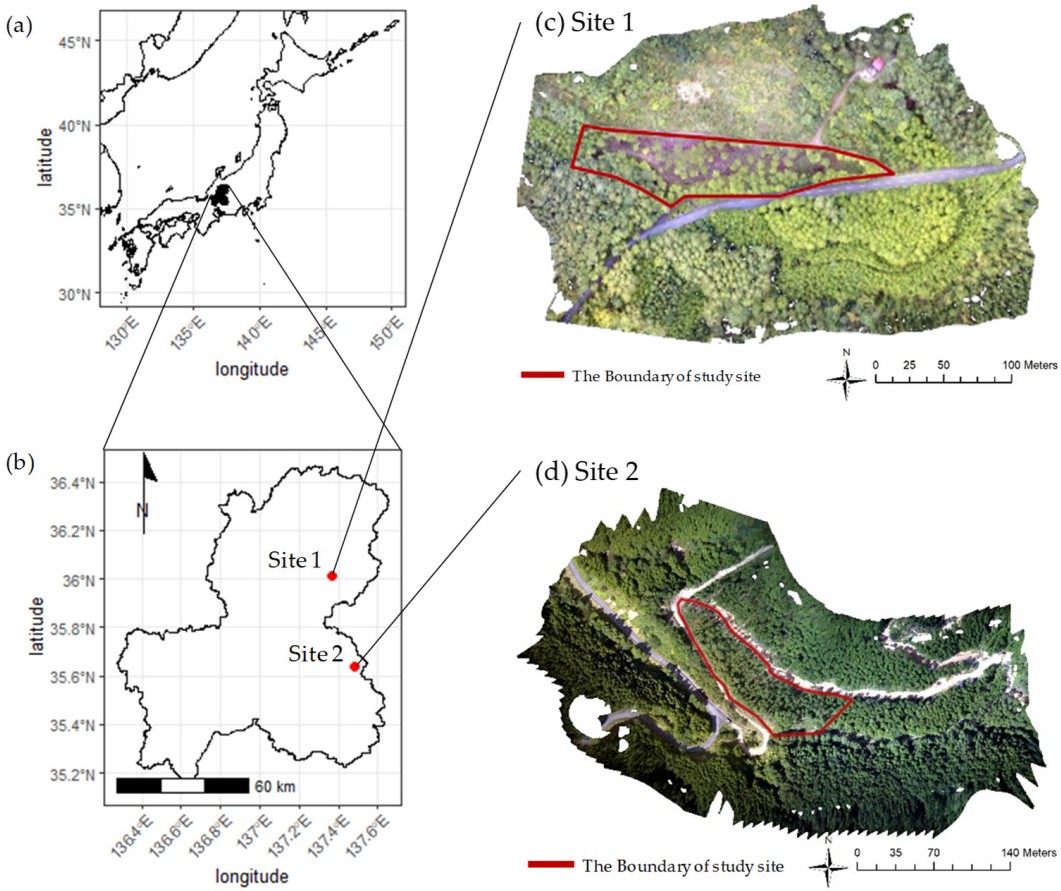

**Figure 2.** Location of the study sites per Fujimoto et al. [6] (Figure 1). (**a**) Gifu prefecture is located in the middle of Japan. (**b**) Red plots represent the locations of two study sites. (**c**,**d**) Orthographic photos of Sites 1 and 2. Red lines show the boundaries of the study sites.

Assuming that the density of woody organs (stems and branches) is constant, the cross section of stems and branches is equivalent to the woody mass per unit vertical length, and the PMT can be formulated using Equation (1) from [23] (Equation (2)).

$$C(z) = \frac{F(z)}{L} \quad (0 \leq z \leq z_0) \, , \tag{1}$$

where $z$ is the vertical distance from the treetop, $C$ is the woody mass (stem and branches) per unit length, $F$ is the cumulative leaf mass from the treetop, $L$ is a parameter named "specific pipe length" with the dimension of length, and $z_0$ denotes the crown height at which an individual tree is divided into an upper (crown) and lower (trunk) part. $L$ is the gradient of the $C \sim F$ relationship and is related to the mechanical strength of the woody organs to support the leaves. From Equation (1), $C$ at crown height $z_0$ is proportional to $F(z_0)$, that is, total leaf mass as Equation (2).

$$C(z_0) = \frac{F(z_0)}{L}. \tag{2}$$

Conversely in a trunk, the total mass (leaves, stems and branches) above any level $z$ is linearly correlated to $C$ at the level. By solving this relationship, Oohata and Shinozaki [25] obtained Equation (3):

$$C(z) = C(z_0) \exp\left(\frac{z-z_0}{a}\right) = \frac{F(z_0)}{L} \exp\left(\frac{z-z_0}{a}\right) \quad (z_0 \leq z \leq H) \, , \tag{3}$$

where $H$ is the tree height and $a$ is a parameter named "specific stress length" with the dimension of length. The aboveground (crown and trunk) woody mass of an individual tree $M$ can be calculated by integrating Equations (1) and (3) from the treetop to the ground as follows:

$$\begin{aligned} M &= \int_0^H C(z)dz = \int_0^{z_0} C(z)dz + \int_{z_0}^H C(z)dz \\ &= \frac{1}{L}\left[\int_0^{z_0} F(z)dz + F(z_0)\int_{z_0}^H \exp\left(\frac{z-z_0}{a}\right)dz\right]. \end{aligned} \tag{4}$$

To utilize a gridded CHM derived by a UAV and segmented into individual tree crowns, leaf mass in a crown was discriminated using CHM cells in an individual tree crown. By assuming that each CHM cell contains an equal amount (area or weight) of leaves, the cumulative leaf mass from the treetop to a certain height is represented as follows:

$$F(z_i) = Almi \quad ( \; i = 1 \sim N, \quad 0 \leq z_i \leq z_N \; ) \, , \tag{5}$$

where $z_i$ is the vertical distance of the $i$-th CHM cell from the treetop, $N$ is the number of CHM cells in a tree crown, $A$ is the area of a CHM cell, $l$ is the leaf area index (LAI; leaf area per CHM cell area) and $m$ is the leaf mass per area (LMA). $z_N$ is the crown height and is identical to $z_0$ in Equations (1) through (4). Using Equation (5), woody mass within a tree crown can be approximated as Equation (6) by applying the trapezoidal rule.

$$\int_0^{z_0} F(z)dz \approx Alm \sum_{i=1}^{N-1} (i+0.5)(z_{i+1}-z_i) = Alm\left[(N-0.5)z_N - \sum_{i=1}^{N-1} z_i\right]. \tag{6}$$

From Equations (4) through (6), the aboveground woody mass $M$ can be calculated using Equation (7).

$$M = A\frac{lm}{L}\left[(N-0.5)z_N - \sum_{i=1}^{N-1} z_i + aN\left\{\exp\left(\frac{H-z_N}{a}\right) - 1\right\}\right], \tag{7}$$

where tree height $H$ can be determined as the highest CHM cell in a crown. Oohata and Shinozaki [25] found that the values of specific stress length $a$ are approximately trunk

height, that is, $H - z_N$ for various forest stands and species. By replacing $a$ with $H - z_N$, Equation (7) can be simplified as follows.

$$
\begin{aligned}
M &= A\frac{lm}{L} \left[ (N - 0.5)z_N - \sum_{i=1}^{N-1} z_i + (H - z_N)N\{\exp(1) - 1\} \right] \\
&= A\frac{lm}{L} \left[ (N - 0.5)z_N - \sum_{i=1}^{N-1} z_i + 1.718N(H - z_N) \right].
\end{aligned}
\tag{8}
$$

The stem volume is proportional to the aboveground woody mass, and we obtained a stem volume estimation model from a UAV-derived crown CHM applying the PMT in Equations (9) and (10).

$$
\begin{aligned}
V = \frac{M}{de} &= A\frac{lm}{deL} \left[ (N - 0.5)z_N - \sum_{i=1}^{N-1} z_i + 1.718N(H - z_N) \right] \\
&= bA \left[ (N - 0.5)z_N - \sum_{i=1}^{N-1} z_i + 1.718N(H - z_N) \right],
\end{aligned}
\tag{9}
$$

$$
b = \frac{lm}{deL},
\tag{10}
$$

where $V$ is the stem volume, $d$ is the volumetric density of wood and $e$ is the biomass expansion factor (BEF) which is the ratio of aboveground mass to stem mass. $b$ is a composition of parameters unique to species and stands, and is the only parameter to be empirically determined in the model.

## 3. Materials and Methods

### 3.1. Study Sites and Field Survey

We used UAV-observed aerial photos and on-site survey results in two forest stands to examine the performance of the stem biomass estimation model. Both sites were mature coniferous tree plantations in Gifu Prefecture, located in the center of Japan Mail Island. Site 1 (36.01° N, 137.37° E and 1000 m a.s.l.) was a 103-year-old Japanese cypress (*Chamaecyparis obtusa* Endl.) plantation covering 0.81 ha. Site 2 (35.64° N, 137.48° E and 680 m a.s.l.) was a 47-year-old Japanese cedar (*Cryptomeria japonica* D. Don) plantation covering 0.96 ha. The two sites partially contained non-dominant species of Japanese cedar and Japanese cypress. Figure 2 shows the locations of the study stands.

The diameter at breast height (DBH), tree height, geographic coordinates, and species were measured for selected individual trees with a DBH $\geq$ 0.05 m or more on 1 November 2016 at site 1, and 31 October and 1 November 2017 at site 2. Diameters (Atsuta Shizai Co., Ltd, Nagoya, Japan) were used to measure the DBH (at 1.3 m above the ground), and tree height poles (SK reverse scale inspection pole, AT-15, Senshin Industry Co., Ltd., Osaka, Japan) and a laser ranging equipment (Laser 550AS, Nikon Imaging Japan Inc., Tokyo, Japan) were used for measurement of the tree height [6].

In the forest survey, we measured the geographic coordinates of 159 and 228 trees at sites 1 and 2, respectively. Of these, we measured the DBH and tree height of 27 Japanese cypress and 5 Japanese cedar trees at site 1, and 15 Japanese cypress and 68 Japanese cedar trees at site 2. The mean height, mean DBH, and tree density of site 1 were 23.8 m, 0.310 m, and 169 ha$^{-1}$, respectively. Those of site 2 were 21.1 m, 0.276 m, and 218 ha$^{-1}$, respectively. All measured trees were used to evaluate the performance of treetop detection and the trees for which DBH and height were measured were used to evaluate the performance of the tree biomass estimation model.

### 3.2. Aerial Photography and Structure from Motion (SfM) Processing

Aerial photography using a UAV was conducted on 21 September 2016 and 31 August 2017 at sites 1 and 2, respectively. A DJI Phantom 3 Professional camera with 12 M pixels and a lens with a focal length of 20 mm (35 mm format equivalent) were used on the UAV for the survey of Site 1, and a DJI Phantom 4 Pro camera with 20 M pixels and a lens with a

focal length of 24 mm (35 mm format equivalent) was used for the survey of Site 2 [6]. The photos were processed using the SfM technique with Photoscan Professional 1.2.6 software (Agisoft LLC, St. Petersburg, Russia) [27], and a 3D point cloud was generated for each site. At site 1, a 3D point cloud containing approximately 44 million points was generated from 129 aerial photos using the SfM technique. At site2, approximately 16 million points were generated from 152 aerial photos.

### 3.3. Canopy Height Model (CHM) Generation, Treetop Detection and Crown Segmentation

The new tree biomass model utilizes the CHM of an individual tree crown. The process of generating individual CHMs from a 3D point cloud observed by a UAV consists of stand CHM generation, individual treetop detection and individual tree crown segmentation. The workflow of this process was the same as that in a previous study [6]; however, the methods in each step were reformed to improve the quality of the derived individual CHMs. All processes were performed using R statistics (Version 4.0.3, Vienna, Austria) [28].

In the first step, stand CHMs with a horizontal resolution of 0.1 m $\times$ 0.1 m were generated from 3D point clouds. In our previous work, we determined the value of a CHM cell from the highest point in each cell, and such a point was sometimes located at the tips of small twigs. As defined in Equation (5), the new biomass model assumes that CHM cells are filled with leaves; therefore, the height around which the points are concentrated is more suitable as the representative value of a CHM cell. In this work, we defined a CHM value as the height at which the maximum point density is reached. The grid_metrics function provided in the lidR package 3.0.4 [29,30] was used to generate CHMs in R statistics, where the point density distribution in a cell was calculated by a Gaussian kernel density function with a window size of 0.1 m.

In the next step, treetops were detected on the stand CHMs using the local maxima filter (LMF) algorithm in a moving window. The number of detected treetops varies depending on the shape and size of the moving window; therefore, their selection rule is crucial for its accuracy. We previously proposed a practical rule to determine the optimum window size as the tipping point of bilinear regression lines on the window size–detection number plot. In our previous work, we used a square moving window and a semi-logarithmic plot (normal scale for window size and logarithmic scale for detection number). In this work, we used a circular moving window considering the isotropic arrangement of treetop locations, and used a double logarithmic plot considering the fractal geometric characters of tree crowns and a forest stand (this theory is discussed in Appendix A). We used the find_trees function with the LMF function in the lidR package of R statistics in this step.

In the last step, we generated the CHMs of individual tree crowns by segmenting the stand CHMs. We used the watershed method to classify stand CHM cells into individual trees similar to a previous study [6]. The only improvement from the previous work was to introduce two criteria to reduce unclassified cells and to prevent misdetection of objects on the ground: the upper distance limit from fragmented cells to crowns to join (set to 2-cell-size), and the lower limit of target cell height defined by the fraction to tree height (set to 0.3).

### 3.4. Tree Biomass Calculation and Model Validation

The stem volume of individual trees in the two sites was calculated using Equation (9) with the segmented crown CHMs, where, cell size $A$ was 0.01 m$^2$ and tree height $H$ was determined from the treetop height on the CHMs. To validate the biomass estimation model, the estimated stem volume was compared with the reference stem volume calculated from the measured DBH and height in the survey using a stem volume model for coniferous species [31]:

$$V = \frac{\pi D^2 H}{4} \left[ 2 \left( 1 - \frac{H_b}{H} \right) \right]^{1.06}, \tag{11}$$

where $V$ is stem volume, $D$ is DBH, $H$ is tree height and $H_b$ is breast height (1.3 m).

## 4. Results and Discussion

### 4.1. Treetop Detection Accuracy

The number of detections monotonously decreases with the size of the moving window when using the LMF to detect treetops. As shown in our previous study, this relationship could be approximated using bilinear regression lines and the tipping point between lines is assumed to be the optimal window size. The improvement in this study was the use of a double logarithmic plot instead of the semi-logarithmic plot in the previous study, and the fitness of the regression lines was improved, as shown in Figure 3. The optimal window diameter to detect treetops was then determined from the tipping points to be 2.4 and 2.0 m in sites 1 and 2, respectively. The area of the circular window was the same as that of the square window in the previous work (2.1 m square) at site 1; however, the area of the circular window was larger than that of the square window (1.5 m square) at site 2. The slopes of the regression lines in the smaller window diameters was –1.93 and –1.52, whereas, that in the larger window diameters was –0.77 and –0.61 in sites 1 and 2, respectively.

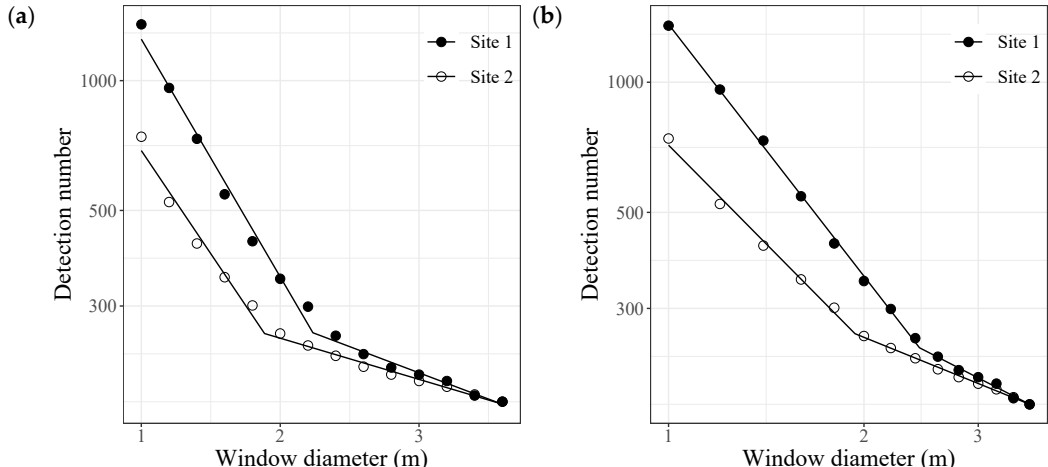

**Figure 3.** Number of detected treetops against the diameter of the moving window at sites 1 and 2. The tipping points between the bilinear regression lines are the optimal widow size. (**a**) is in a semi-logarithmic scale (normal scale for window diameter), and (**b**) is in a double logarithmic scale.

A total of 223 and 259 treetops were detected in sites 1 and 2, respectively. The coverage of the survey was not complete and numerous trees were left unmeasured because of access difficulty as well as the screening of small trees (DBH less than 0.05 m). In our previous study, only true positives (TPs) and false negatives (FNs) were assessed because false positives (FPs) were unknown; however, reducing FNs by using a smaller window size always involves a tradeoff against increasing FPs. By assuming that the measured trees in the survey were perfect at site 2 where the coverage of the survey was rather high, precision (TP/(TP + FP)), recall (TP/(TP + FN)) and F score (2 × precision × recall/(precision + recall)) were calculated as shown in Figure 4 against the moving window diameter. Precision clearly increased with window diameter, whereas recall displayed an opposite trend. The F score showed a broad inversed U shape and peeked at 0.78 with the window diameter of 2.0 m. This window size was the same as that determined by the tipping point of the bilinear regression on a double logarithmic plot, which was a larger F score than that at the optimal window size in our previous work (0.76 at a diameter of 1.7 m). Thus, the window size selection method in this study provides an improved overall accuracy compared with that with the previous method even though recall was lower (0.83 and 0.92 at the window sizes in this and previous studies, respectively). If the coverage of the survey is perfect, less FPs are expected (precision must be higher), so the actual overall accuracy for treetop detection would be improved compared to that of the above results.

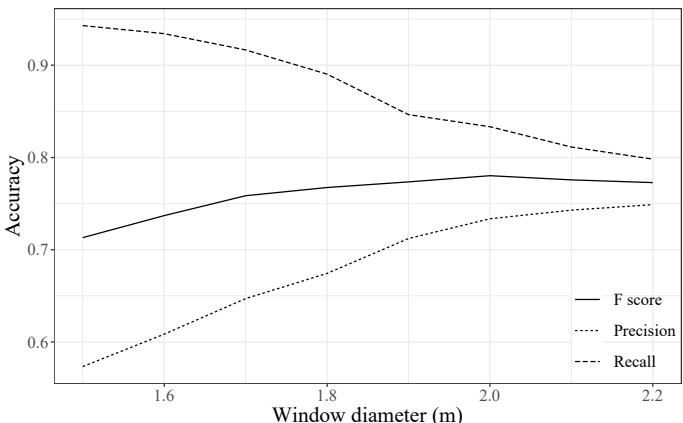

**Figure 4.** Accuracy of treetop detection with different moving window diameters of the local maxima filter (LMF) in site 2. Precision is $TP/(TP + FP)$, recall is $TP/(TP + FN)$ and F score is $2 \times \text{precision} \times \text{recall}/(\text{precision} + \text{recall})$, where, TP, FP and FN denote true positive, false positive and false negative, respectively.

Many studies have employed the LMF algorithm for treetop detection in CHMs [32–35]; among them, some selected the optimal window size by trial and error and others were "empirically" determined or did not state the method. The reported treetop detection accuracy of these previous studies was 0.91–0.96 for recall [32], 0.94 for recall [33]; 0.86 for F score [34]; 0.82 for F score [35]. The recall and F scores in this study were lower than those reported; however, considering that the coverage of the survey was not perfect, the accuracy of the treetop detection in this study can be considered acceptable. Therefore, if surveys of reference trees are not available, the proposed method to determine the optimal window size of LMF using only CHMs is useful and reasonable.

Finally, the CHMs segmented into individual tree crowns were derived as shown in Figure 5. Small trees with a crown size less than 1 m$^2$ were eliminated. As a result, the number of individual CHMs was 216 and 238 in sites 1 and 2, respectively. Among them, 30 and 76 trees with DBH and tree height measurements from the survey were used to validate the biomass model, respectively.

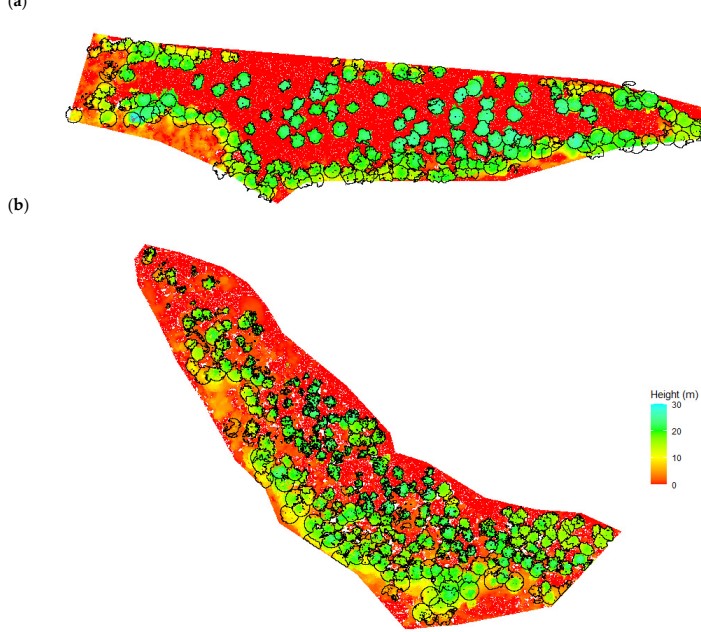

**Figure 5.** Canopy height models segmented to individual tree crowns at sites (**a**) 1 and (**b**) 2.

### 4.2. Trimming Individual Tree CHMs Using the PMT

It was found that individual tree CHMs after segmentation by the watershed method were occasionally deformed from the general shape of the crown of coniferous species. An example is shown in Figure 6a, where extra leaves are distributed at the lower positions of a crown separately from the main leaves at the higher position. The lower leaves in Figure 6a were from a neighboring short tree whose treetop had not been detected. Possible reasons of these extra leaves are that objects on the ground and dead branches at the lower position were observed in the aerial photography, that points could have been mis-located in the SfM process because of the low visibility at the lower positions, and that the leaves of neighboring tree crowns could have been mis-segmented. These extra leaves at the lower positions should be trimmed from individual CHMs before being used in the tree biomass estimation model.

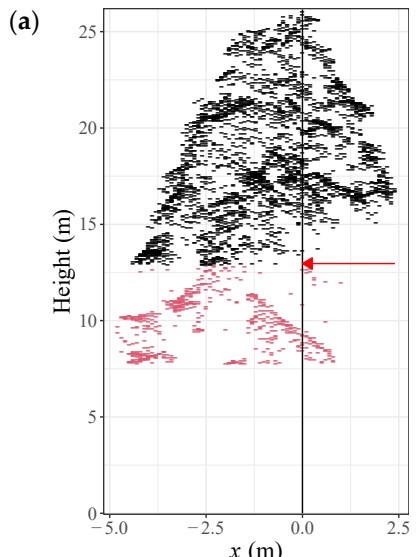
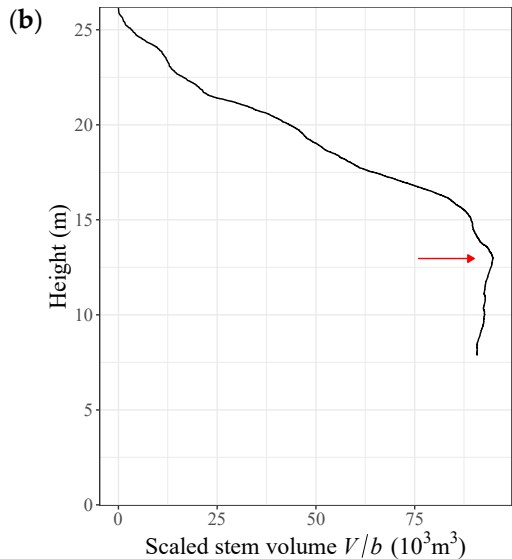

**Figure 6.** (**a**) Example of a tree shape segmented from a canopy height model by the watershed method. (**b**) Scaled stem volume $V/b$ change by height of the individual in (**a**) calculated by Equation (9). The red arrows represent the height at which $V/b$ reaches its maximum, and red leaves on (**a**) are the leaves below the height of the arrows.

The canopy structure theory of Monsi and Saeki [36], which serves as the methodological base of the PMT, analytically showed the vertical mass distribution of assimilating and non-assimilating organs in a plant canopy, and suggested that the optimal leaf area index is autonomously determined to maximize the net production of the whole plant canopy. From an analogy to the canopy structure theory, the PMT can be used to distinguish the extra leaves in a CHM by searching for the optimum $z_N$ to maximize woody biomass. Let $V/b$ be a scaled stem volume from Equation (9), which is plotted against $z_i$ in Figure 6b. $V/b$ increased from the treetop as $z_i$, reached its maximum at $z_i = 13.18$ m (12.96 m above ground), and then decreased up to $z_N$, which indicated that the leaves below the optimal $z_i$ do not contribute to biomass growth and can be eliminated. Thus, the optimal $z_i$ and corresponding cell order from the treetop $i$ could be new $z_N$ and $N$, respectively, and the cells below the new $z_N$ were trimmed out from individual tree CHMs. The new crown height $z_N$ was shorter than that before trimming by an average of 7.70 m and a maximum of 17.05 m, and the new canopy area $AN$ was smaller by an average of 1.44 m$^2$ and a maximum of 11.06 m$^2$.

### 4.3. Implementing the Density Effect in the Tree Biomass Model Applying the PMT

The tree biomass estimation model applying the PMT has only one parameter to be determined empirically (parameter $b$ in Equations (9) and (10)). The average $b$ calculated from Equation (9) and the reference stem volume using Equation (11) with measured

DBH and tree height $0.28 \times 10^{-5}$ for all species and sites and which widely varied from $0.09 \times 10^{-5}$ to $2.27 \times 10^{-4}$; therefore, it was not possible to obtain a constant parameter. The variation in $b$ could not be explained by the species and sites according to the analysis of variance. As shown in Equation (10), parameter $b$ is the composed one of five parameters, and was assumed to be species- and site-specific. However, some of the component parameters may vary according to the individual trees. The most likely effect of changing parameters is canopy crowdedness, which is well known as the density effect. Generally, in a dense tree canopy, the crown is small and stems are thin [37]. The sensitivity of each component parameter to canopy crowdedness is unknown, but the smaller biomass expansion factor $e$ and specific pipe length $L$ in a dense canopy are reasonable.

Following the discussion above, we examined a variety of indicators representing canopy crowdedness, and selected the aspect ratio of a tree crown (ratio of crown height to crown projected area) as the key factor determining the variety of $b$. Finally, we revised the tree biomass estimation model by applying the PMT as follows:

$$
\begin{aligned}
V \quad &= b\prime \tfrac{z_N}{NA} A \left[ (N - 0.5)z_N - \sum_{i=1}^{N-1} z_i + 1.718N(H - z_N) \right] \\
&= b\prime \tfrac{z_N}{N} \left[ (N - 0.5)z_N - \sum_{i=1}^{N-1} z_i + 1.718N(H - z_N) \right],
\end{aligned}
\tag{12}
$$

$$
b\prime = \frac{NA}{z_N} b,
\tag{13}
$$

where $b\prime$ is a new integrated parameter to be determined empirically with the dimension of length.

### 4.4. Stem Volume Estimation by the Tree Biomass Model Applying the PMT and Its Validation

Prior to validating the model, the accuracy of tree height estimated using UAV-SfM was assessed. The maximum error, mean error and root mean square error (RMSE) of tree height were 4.17 m, –0.50 m and 1.21 m, respectively. The RMSE was larger than that of successful results, for example, 0.479 m reported in a comparative study in which the UAV-SfM technique was used [38]. A part of the under estimation in tree height was due to the cell height determination of a CHM in which the height at the maximum point density in a cell was selected instead of the highest point, and this rule reduced CHM height by an average of 0.3 m. However, a considerable gap remained between the estimated and measured tree heights. The other possible causes are the quality of the aerial photos (lighting conditions affect the point generation performance), precision of the optical devices, the performances of the SfM software and hardware (precision depends on the amount of computation), accuracy of the field survey that measured tree height using a laser ranging equipment and a tree height pole. The accuracy of the tree height is critical in tree biomass estimation in any model scheme using remote-sensing techniques and should be improved.

The estimated and reference stem volumes are shown in Figure 7, and estimation accuracy metrics and parameter $b\prime$ are summarized in Table 1, where the results of the subsets classified by species and sites are also shown. The stem volume RMSE was 0.26 m$^3$ for all samples and it was equivalent to a relative error of 32% for the average stem volume. The RMSE of the subsets by species and sites ranged from 0.24 m$^3$ to 0.29 m$^3$ and did not vary greatly among species and sites. The value of $b\prime$ was $1.28 \times 10^{-3}$ m for all samples, and varied from $1.22 \times 10^{-3}$ m to $1.33 \times 10^{-3}$ m by species and sites, of which the range was within $\pm 5\%$ and was not large. The $b\prime$ of Japanese cypress was smaller than that of Japanese cedar, which is consistent with the fact that the volumetric density of wood $d$ of Japanese cypress is larger than that of Japanese cedar (see Equation (10)).

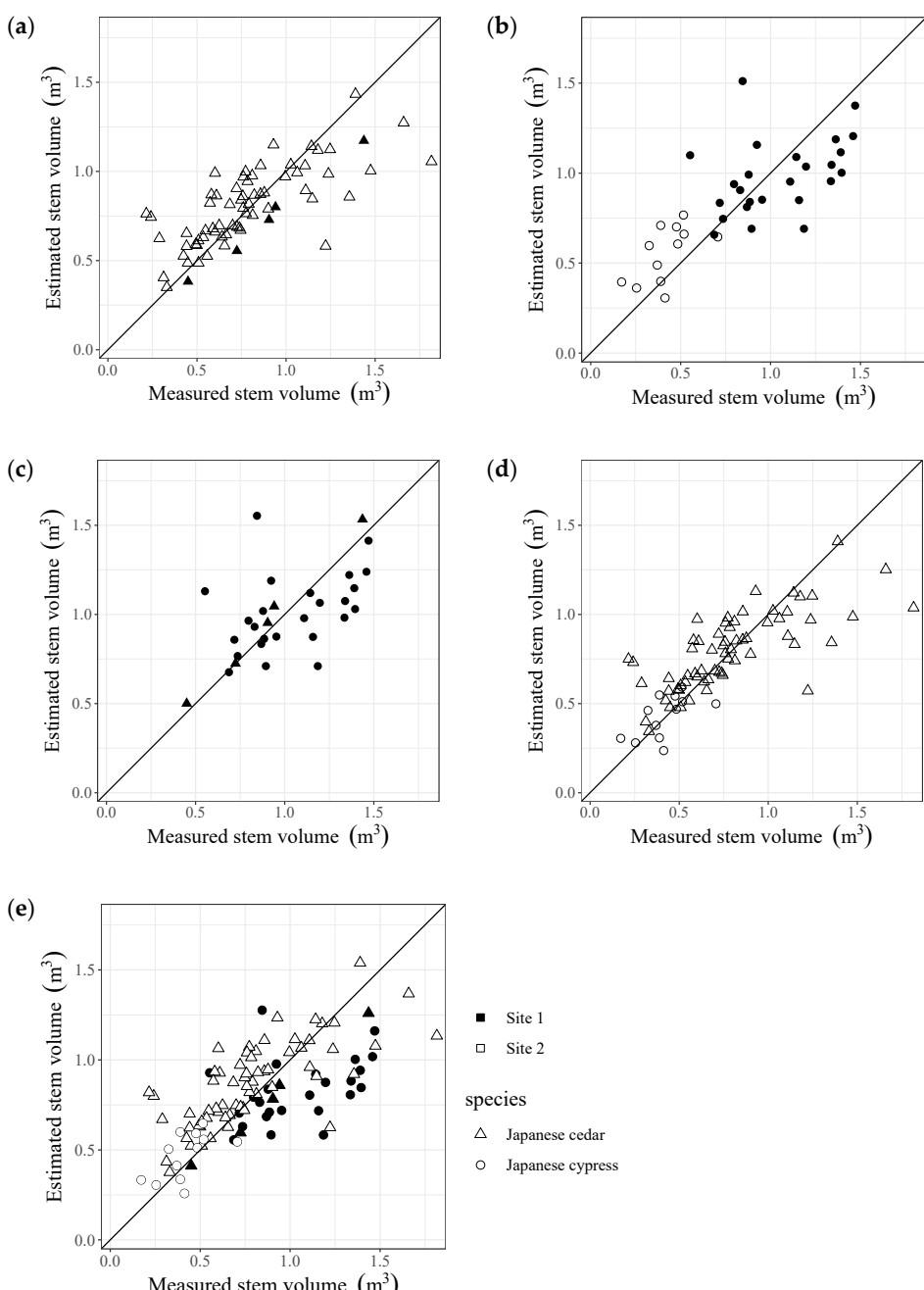

**Figure 7.** Comparison between the reference tree volume and those estimated using the tree biomass estimation model applying the pipe model theory. (**a**) and (**b**) are Japanese cedar and Japanese cypress in all sites, respectively. (**c**) and (**d**) are all species in sites 1 and 2, respectively. (**e**) is all species in all sites.

**Table 1.** Accuracy of tree stem volume estimation using the tree biomass estimation model applying the pipe model theory.

| Dataset | Samples | RMSE (m³) | $R^2$ | $b'$ ($10^{-3}$ m) |
|---|---|---|---|---|
| Japanese cedar in all sites | 69 | 0.27 | 0.37 | $1.33 \pm 0.05$ |
| Japanese cypress in all sites | 37 | 0.24 | 0.59 | $1.22 \pm 0.05$ |
| All species in site 1 | 30 | 0.29 | 0.17 | $1.26 \pm 0.06$ |
| All species in site 2 | 76 | 0.25 | 0.46 | $1.30 \pm 0.05$ |
| All species in all sites | 106 | 0.26 | 0.45 | $1.28 \pm 0.04$ |

The accuracy of the new tree biomass estimation model applying the PMT was compared with that of other methods. One method used the DBH estimation model [19] combined with Equation (11), which was presented in our previous study [6]. The DBH model equation is as follows:

$$D = f \times 0.557(H \times CD)^{0.809} \exp\left(\frac{0.056^2}{2}\right), \tag{14}$$

where $D$ is DBH (cm), $H$ is tree height (m), $CD$ is crown diameter (m), and $f$ is a correction factor for the study sites introduced only in this study. As a result, DBH was estimated with an RMSE of 0.05 m; nevertheless, the estimated stem volume considerably underestimated (see Figure 8a) with an RMSE of 0.34 m$^3$ and an $R^2$ of 0.66. This comparison suggests the high sensitivity of tree biomass estimation to DBH accuracy and the advantage of the new model which does not require DBH as an intermediate parameter for tree biomass.

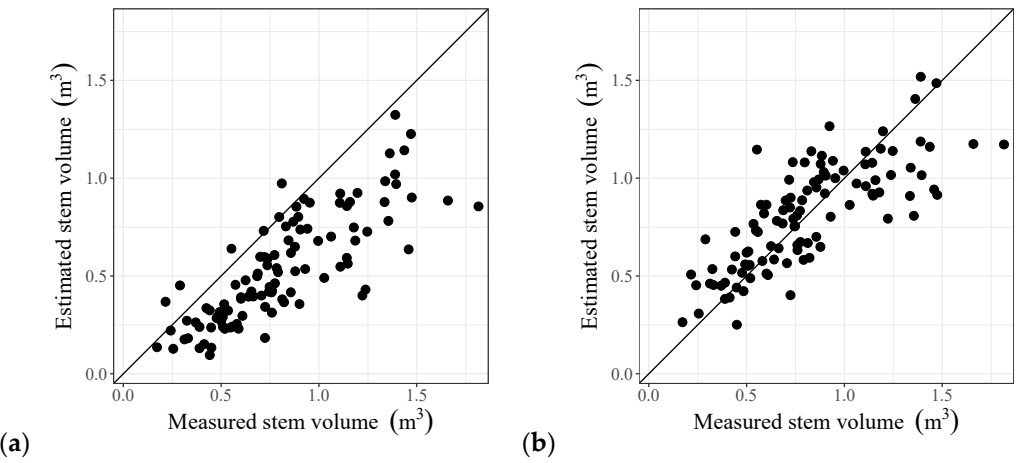

**Figure 8.** Comparison between the reference tree volume and those estimated using the two comparative models: (**a**) the DBH estimation model [19] and (**b**) the non-linear multiple regression model [21].

Another comparative stem volume estimation method is an empirical model by Itoh et al. [21] that examines a variety of formulas and tree canopy metrics to estimate stem volume from LiDAR measurements, and proposes the following equation as the best model for Japanese cedar.

$$V = 10^{\alpha} H^{\beta} As^{\gamma}, \tag{15}$$

where $\alpha$, $\beta$ and $\gamma$ are empirical parameters, $H$ is tree height and $As$ is crown surface area, which was calculated from the horizontal area and inclination of CHM cells in this study. The RMSE of stem volume was 0.23 m$^3$ and the $R^2$ was 0.60, and was slightly improved compared to that of the new model (see Figure 8b). As mentioned in Section 1, multivariate non-linear empirical models can potentially achieve high accuracy; however, attention should be paid to overfitting and parameter generalization. In this sense, the tree biomass estimation model applying the PMT has the advantage of generalization to other types of forest because it contains only one adjustable parameter and its variability was shown to be rather small among species and sites.

### 4.5. Future Perspectives on the Tree Biomass Estimation Model and Processes

Although the tree biomass model applying the PMT proposed in this study achieved a comparative performance to that of previous models in the study sites and for two species, several perspectives should be studied to improve the model usage as well as the processing method. Regarding the model usage, first, the stability of model parameter $b\prime$ is not clear. $b'$ was shown to be less variable in this study; however, its stability to

various species including deciduous or mixed species and site characteristics such as age, stem density, and complex terrain are still unknown. Second, this study only considered the density effect of the canopy to correct parameter *b*; however, the variability of other parameters composing *b* has not yet been assessed. Examples are the variabilities of LAI *l* by species and site condition, LMA *m* by species and leaf position (sunlit or shaded), and specific pipe length *L* by species, site, and age (because it reflects wood strength).

Regarding the data-processing method to generate individual tree CHMs for use in the model, first, the treetop detection algorithm should be improved to achieve higher accuracy. The optimal size of the moving window used with the LMF for treetop detection was determined based on fractal dimensions in this study, and it was fixed at each site. The crown size of individual trees varies within one stand; therefore, the optimal window size for treetop detection must differ according to the location within a stand. Previous studies examined a variable window size of LMF according to height [39]; however, in crowded stands, crown size is also sensitive to stem density [37]. More effective algorithms with variable window sizes need to be developed to improve the treetop detection performance. Second, a simple watershed method to segment a stand CHM into individual trees may not be effective for dense deciduous forests whose canopy is continuous and the crowns of individuals overlap. In such stands, segmentation by only surface height would be difficult and inaccurate; therefore, the use of additional information such as colors and brightness is worth examining.

The proposed tree biomass estimation model in this study used individual tree CHMs as input data and assumed that the cells of CHMs contain equal amounts of leaves and that there were no leaves beneath the cells, which was similar to a shelter. This assumption is reasonable for species whose leaves are concentrated on the crown surface, such as Japanese cedar, but may not be suitable for species whose leaves are partially distributed inside crowns, such as Japanese cypress and deciduous species. The points in a CHM cell from a point cloud generated using the UAV-SfM technique are also distributed beneath the height of the cell surface, but they cannot be used as an indicator of leaf density because of the low positioning accuracy and uneven point generation rate at the lower positions. Recently, UAV-borne LiDAR systems have been developed and utilized in both research and forest management practices. The advantages of LiDAR compared with aerial photography are high vertical distance accuracy and the ability to provide a vertical profile of leaf density. The tree biomass estimation model is potentially capable of using vertically distributed leaf density instead of the canopy surface height of CHMs by transforming the constant LAI *l* × LMA *m* in Equations (5) through (10) to a variable that changes with the distance from treetop $z_i$.

## 5. Conclusions

To develop a new tree biomass estimation model adaptable to airborne observations of the forest canopy by unmanned aerial vehicles (UAVs), we excavated old allometric theories of plant form, the pipe model theory (PMT), and the statical model of plant form as an extension of the PMT for tall trees. The achievements and findings of this study are summarized below.

- The PMT is adaptable to the tree biomass estimation using UAV-derived CHMs because it can provide the tree form without using DBH.
- The optimal moving window size used in the local maxima filter (LMF) algorithm can be determined as the tipping point of the bilinear regression lines on a double logarithmic plot between window size and detection number.
- The PMT is also applicable to distinguish the misclassified extra leaves at lower positions in an individual tree CHM.
- The stem volume estimation accuracy of the new model was greater than that of a DBH estimation model and was comparable to that of a three-parameter empirical model.
- An advantage of the new model in generalization is that it contains only one parameter to be empirically adjusted. This parameter is rather stable in different species and sites.

- The new model is applicable to the vertical profile data of leaf density observed by airborne LiDARs.

Further applicability and accuracy assessment as well as characterization of the empirical parameter are needed to confirm and improve the validity of this new model to a variety of forest stands.

**Author Contributions:** T.M. was responsible for conceptualizing the model and methodology, data processing and analysis, interpretation and discussion, and A.F. was responsible for a prototype work that was succeeded by this study. K.H., S.S. and H.T. conducted UAV handling and 3D point cloud generation. All the authors conducted the field survey. All the authors contributed to the revision of the manuscript. All authors have read and agreed to the published version of the manuscript.

**Funding:** This study was supported by the Joint Usage/Research Program of the Institute of Materials and Systems for Sustainability (IMaSS), Nagoya University, and by the Collaboration Research Program of IDEAS, Chubu University IDEAS201609.

**Acknowledgments:** The authors thank Chubu Forest Co. for providing us the opportunity to study the sites.

**Conflicts of Interest:** The authors declare no conflict of interest.

## Appendix A Fractal Geometry of Tree Crowns and Forest Stands

Treetops were detected in canopy height models by using the local maxima filter (LMF) algorithm with a moving window. As shown in Section 3.2., the number of detected treetops varies depending on the size of the moving window, and its selection is crucially important for accuracy. This study proposed the optimal window size as the tipping point of the bi-linear regression lines on a double logarithmic plot of window size and detection number, and its performance is shown in Section 4.2. This appendix discusses the theoretical rationality of the proposed method.

Since the introduction of fractal geometry [40], it has had a major impact on modeling and analysis in the natural and physical sciences [41]. Many studies have reported the fractal-like structures of tree crowns and forest canopies. The fractal dimension is a key indicator of the structural characteristics of natural objects. For example, a volume-filling foliage has an Euclidian dimension of 3; however, the fractal dimension of tree crowns is actually less than 3 because foliage concentrates near the crown surface due to the shading effect [42]. At the stand level, the scaling exponents between tree size and number or canopy topography have been discussed in relation to self-thinning and local site quality. [43–45]. Examples of fractal dimensions (including the scaling exponent) of tree crowns in previous studies were 2.64 [42], 2.2 [46], and 2.24–2.45 [47], and those of stands were 1.71 [48], 1.78–1.94 [49], and 1.92–1.95 [50], showing that the fractal dimension of crowns is between 2 and 3 and that of stands is between 1 and 2.

The box-counting analysis is an appropriate method of fractal dimension estimation for images [51], in which, the fractal dimension $Df$ can be determined as the slope of the number of boxes containing objects in an image $N(s)$ versus box size $s$ on a logarithmic scale as follows:

$$\log N(s) = -Df \log s + K, \tag{A1}$$

where $K$ is the intercept. This relationship is essentially equivalent to that between the treetop detection number and window size of the LMF on a double logarithmic plot, as shown in Figure 3. The slope in Figure 3 was –1.93 and –1.52 in the smaller window diameters, and was –0.77 and –0.61 in the larger window diameters in sites 1 and 2, respectively. Therefore, these slopes can be said to represent the fractal dimensions of crowns and stands, respectively. Note that the fractal dimension examples shown above were from 3D observations, and they would be reduced by 1 if using the projected objects on a horizontal plane, similar to the treetop point raster images. Therefore, it can be said that the tipping point between the bilinear regression lines on a double logarithmic plot

of window size and detection number indicates the threshold scale between crowns and stands. Thus, it is optimal for treetop detection using the LMF algorithm.

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
