# Peer review of "A Novel Tree Biomass Estimation Model Applying the Pipe Model Theory and Adaptable to UAV-Derived Canopy Height Models"

_forests, doi:10.3390/f12020258_

Round 1

Reviewer 1 Report

Dear Authors,

Please find the comments attached.

Regards

Author Response

Dear reviewer,

The authors sincerely give thanks to the reviewer who has kindly provided precise and valuable advices to help the improvement of the quality of our manuscript. We have revised the manuscript following your advises, and we answer to all the comments. Please find in the notes of the PDF file.

Reviewer 2 Report

The manuscript was aimed to develop a new tree biomass estimation model adopted to UAV observations of forest canopies based on the pipe model theory.

The topic of the study is interesting and of current interest. The methodology described has a scientific background and novelty. The manuscript is well written and has a good understandable structure.

However, there are several comments to be considered to improve the manuscript:

Line 69: these three approaches estimate biomass from estimates, not from measurements in the field.

Line 109: woody organs – is this the same as woody mass (stem and brunches)?

Line 151, 166: I would recommend to mention here about type of the camera used (RGB?) and spatial resolution.

Line 215, 224-229: the section about field measurements is absent. It is recommended to include the information to the separate section about field data.

Line 244: Figure 3, right graph, axis X. The distance between 1 and 2 does not correspond to the distance between 2 and 3.

Line 323. Description of Figure 6. ..and leaves below the height? Or below the canopy?

Line 338. As an example to compare your results with other studies I would recommend Marzulli et al. 2019, https://doi.org/10.1093/forestry/cpz067, Krauss et al. 2019, Goodbody et al. 2016 https://doi.org/10.1080/01431161.2016.1219425.

Line 436-439: Information about UAV-Lidar systems can be moved to the Intro.

Line 463: potentially could be applicable.

Author Response

Dear Reviewer,

The authors sincerely give thanks to the reviewer who has kindly provided precise and valuable advices to help the improvement of the quality of our manuscript. We have revised the manuscript following your advises, and we answer to all  the comments. Please find in the response letter.
